# Synergistic Use of Sentinel-1 and Sentinel-2 Based on Different Preprocessing for Predicting Forest Aboveground Biomass

Gengsheng Fang [1,2], Hangyuan Yu [3], Luming Fang [1,2] and Xinyu Zheng [1,2,*]

1   College of Mathematics and Computer Science, Zhejiang A & F University, Hangzhou 311300, China
2   Key Laboratory of Forestry Intelligent Monitoring and Information Technology Research of Zhejiang Province, Zhejiang A & F University, Hangzhou 311300, China
3   Qianjiangyuan National Park Management Bureau, Quzhou 324000, China
*   Correspondence: zhengxinyu@zafu.edu.cn

**Abstract:** Forest aboveground biomass (AGB, Mg/ha) measurement is one of the key indicators for carbon storage evaluation. Remote sensing techniques have been widely employed to predict forest AGB. However, little attention has been paid to the implications involved in the preprocessing of satellite data. In this work, considering the areas of low forest AGB in our survey plots, we explored the implications of employing atmospheric correction and speckle filtering with Sentinel-1 (S1) synthetic aperture radar (SAR) and Sentinel-2 (S2) to predict forest AGB using multiple linear regression (MLR) and extreme gradient boosting (XGBoost). In the present study, the types of plots examined included oaks (*Quercus* spp.), Chinese firs (*Cunninghamia lanceolata*), and Masson pines (*Pinus massoniana*), and all of the plots were investigated. Specifically, the feature variables related to S1 (dual polarization and texture measures) and S2 (spectral bands) were modeled individually, and 16 feature sets, including different combinations of S1 and S2 based on different preprocessing measures, were established using MLR and XGBoost. The results show that speckle filtering and atmospheric correction marginally influenced the capacity of the S2 spectral bands, the SAR dual-polarization backscatter, and the SAR-based textural measures in predicting the AGB in our survey plots. The associations between the speckle-filtered and unfiltered SAR images and the S2 Top-of-Atmosphere and Bottom-of-Atmosphere products were considerably strong. Additionally, the texture models generally showed better performances than the raw SAR data. Ultimately, the groups that only encompassed the S2 spectral bands were the best-performing groups among the 16 feature sets, while the groups that included only S1-based data generally performed the worst.

**Keywords:** AGB; Sentinel-1; Sentinel-2; preprocessing; atmospheric correction; speckle filtering

## 1. Introduction

Forest ecosystems are critical drivers of global climate-change mitigation via their capacity for carbon storage and ingestion [1]. The standardized assessment of forest aboveground biomass (AGB, Mg/ha) plays a vital role in the estimation of carbon fluxes [2] and renewable resource potential [3]. With the evolution of satellite tools, AGB has been extensively evaluated on a large scale by utilizing spatial resolution patterns that vary from high to medium [4] to overcome the troublesome sampling and deferred monitoring issues associated with orthodox measurements [5,6].

With 13 spectral bands, ranging from 10 to 60 m spatial resolutions and accompanied by a five-day revisit cycle [7], the Sentinel-2 (S2) MultiSpectral Instrument (MSI) has been employed in a variety of previous AGB studies [8–10]. Furthermore, boasting medium (typically 10 m) resolutions and a six-day, exact revisit time [11], the Sentinel-1 (S1) C-band synthetic aperture radar (SAR) has also commonly been used in previous studies, especially for studying low forest AGB, such as that in the survey plots considered in this research. Its high temporal and spatial resolution time series [12] and its capability

of penetrating forest canopy layers [13] have allowed Sentinel-1 to make tremendous contributions to vegetation observations [14,15]. Additionally, complementary efforts have been made by integrating optical sensors with SAR so as to exploit both of their strengths [13,16,17]. Nevertheless, it remained unknown as to whether satellite-based images using disparate preprocessing procedures, including atmospheric correction for Sentinel-2, speckle filtering for Sentinel-1, and the integration of Sentinel-1 and Sentinel-2, could distinctly affect estimates of forest AGB.

The atmospheric correction (AC) method refers to the procedure that transforms Top-of-Atmosphere (TOA) Level-1C products into Bottom-of-Atmosphere (BOA) products [18], counteracting the contamination associated with aerosols or water vapor [19]. On the other hand, as discussed in reference [20], supposing that interventions in the atmosphere are minimal, atmospheric correction might not be necessary. Speckle filtering has been applied to cut down multiplicative granular noise generated by SAR [21,22], helping to improve the performance of land-use classification and pattern interpretation [23]. Presently, it remains unknown as to whether AC and speckle filtering can greatly influence predictions of forest AGB that rely on S1 and S2 images.

Texture features refer to the spatial heterogeneity delivered via the gray-tone values of pixels in a given sliding-window size with respect to image interpretation [24,25]. As popular and promising predictors in forest biomass research [25–27], image textures are the local variance in gray tones, and depend on the forested scope and image resolution [25,28]. Textural metrics have probably been considered as having greater potential than original spectral parameters, especially when processing remotely sensed images [29,30]. Furthermore, methods of image texturing have been developed into first-order statistics (occurrence) and second-order statistics (co-occurrence) [31,32]. First-order textures are calculated from the histogram of pixel intensities in a pattern, ignoring the interaction between neighboring pixels, while second-order textures are based on the gray-level co-occurrence matrix (GLCM), shedding light on spatial permutations yielded by the juxtaposition of specified gray-tone values [32,33]. Both first-order and second-order texture measures have been extensively applied to the prediction of sundry forest variables [34–36]. Nevertheless, speckle filtering is considered to be a concession between speckle elimination and fine-detail preservation, with the compromise being that it probably mistakenly blurs texture measures to some extent [37]. In addition, reference [38] reported that the speckle-filtering approach counteracts the employment of backscatter textures in SAR images to predict forest inventory characteristics. Thus, we attempt here to determine whether speckle filtering for SAR images plays a crucial role in the prediction of forest AGB.

In order to determine whether the functions of S1 and S2 could make a difference in evaluating forest AGB using different preprocessing techniques, along with the extraordinary capacity of SAR-based textures, as previously reported in [30,39], we focused on the relationships between sensor groups that adopt different preprocessing steps and the AGB of three types of plots, including ones dominated by oak (*Quercus* spp.), Chinese fir (*Cunninghamia lanceolata*), and Masson pine (*Pinus massoniana*), along with all of the survey plots that include a variety of dominant species in Lin'an District (Hangzhou, China). Our research goals encompassed the following: (1) to compare the performance of S2, with and without AC, against S1 SAR and SAR-based texture measures, with and without speckle filtering; (2) to assess the combination of S2, with or without AC, coupled with SAR and two kinds of SAR-based textures (first-order and second-order), with and without speckle filtering; and (3) to investigate the synergistic employment of S2, S1, and S1-based textures via various preconditioning measures.

## 2. Materials and Methods

### 2.1. Study Site

The research area (118°51′~119°52′ E, 29°56′~30°23′ N) is located on the western edge of Hangzhou, Zhejiang Province (118°01′~123°10′ E, 27°06′~31°11′ N) in Southeast China

(Figure 1). With a total area of approximately 312,600.8 ha, the landscape of Lin'an is dominated by medium and low hills.

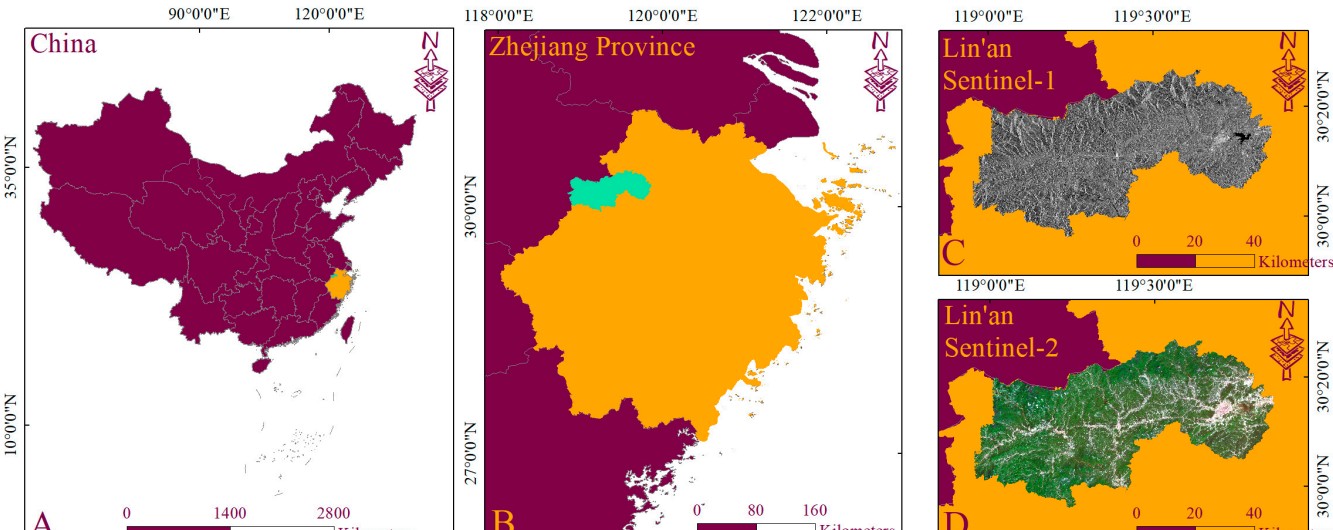

**Figure 1.** (**A**) represents the geographical location of Zhejiang Province in China; (**B**) represents the geographical location of Lin'an District in Zhejiang Province; (**C,D**) represent images of Lin'an District as produced by Sentinel-1 and Sentinel-2, respectively.

Lin'an has a monsoon climate with an average annual rainfall of 1613.9 mm and an average temperature of 16 °C; the area experiences four distinct seasons. The forest types in Lin'an District are mainly broadleaf evergreen forests, coniferous forests, and mixed coniferous and broadleaf forests. Furthermore, the subtropical evergreen–broadleaf forest is the major forest type in this region.

### 2.2. Field Data Source

The forest inventory data for our survey fields were obtained from the Zhejiang Forestry Department, which collected them in 2017. The entire area of the research field is 534.52 km². In our study, the research plots were set by taking forest resource surveys and management into account. With the help of the intersection of kilometer networks referring to the topographic maps with a 1:10,000 map scale and National Forest Inventory standards, the research plots were divided using Arcgis, according to terrain boundaries (e.g., mountain ridges, valleys, roads, etc.) and similarities among the tree species. The research plots were also summarized by their dominant tree species, which refers to the tree species with a proportion of growing stock volume (GSV, m³/ha) equal to or greater than 65% of the total GSV in a plot. The overall number of plots in Lin'an, Zhejiang Province, is 119,793, and 20,690 plots, ranging in area from 0.05 to 36 ha, were considered in this work. The survey data acquired in the field were based on stratified random sampling with the help of remotely sensed data with a spatial resolution greater than 2.5 m (most were detailed using high-resolution unmanned aerial vehicles (UAV)).

Three dominant tree species, the Chinese fir (*Cunninghamia lanceolata*), the Masson pine (*Pinus massoniana*), and the oak (*Quercus* spp.), were identified in this study. Furthermore, all of the plots that included one of the above-mentioned species as the dominant species were considered in our work. The census of the dominant tree species is displayed in Table 1. Simultaneously, the forest AGB was calculated according to the allometric equations acquired from previous research. In the equations, *D* and *H* represent the diameter at breast height and the tree height, respectively. Since the low penetration capability of the C-band backscatter for low forest AGB plots was more appropriate to studying the low forest AGB in our survey plots, as shown in Table 1, C-band Sentinel-1 was utilized for the following AGB predictions.

**Table 1.** Statistical information about the dominant species in the sampling plots.

| Dominant Species | Allometric Equations | Number of Plots | Forest Variable | Mean | Std |
|---|---|---|---|---|---|
| Oak (*Quercus* spp.) | $0.13188D^{1.82892}H^{0.71119}$ [40] | 5291 | DBH (cm) | 11.56 | 2.70 |
| | | | Tree Height (m) | 7.76 | 1.89 |
| | | | GSV (m³/ha) | 51.99 | 30.80 |
| | | | Age (years) | 27.50 | 8.82 |
| | | | AGB (Mg/ha) | 55.31 | 41.64 |
| Chinese fir (*Cunninghamia lanceolata*) | $0.065388D^{2.01735}H^{0.49425}$ [41] | 8592 | DBH (cm) | 14.25 | 3.11 |
| | | | Tree Height (m) | 10.33 | 2.86 |
| | | | GSV (m³/ha) | 73.49 | 33.39 |
| | | | Age (years) | 22.78 | 7.78 |
| | | | AGB (Mg/ha) | 48.27 | 25.01 |
| Masson pine (*Pinus massoniana*) | $0.066615D^{2.09317}H^{0.49763}$ [42] | 6807 | DBH (cm) | 19.39 | 2.86 |
| | | | Tree Height (m) | 13.76 | 2.02 |
| | | | GSV (m³/ha) | 146.79 | 47.28 |
| | | | Age (years) | 35.37 | 7.50 |
| | | | AGB (Mg/ha) | 126.80 | 48.75 |
| All plots | | 20,690 | DBH (cm) | 15.25 | 4.26 |
| | | | Tree Height (m) | 10.80 | 3.31 |
| | | | GSV (m³/ha) | 92.11 | 54.59 |
| | | | Age (years) | 28.13 | 9.63 |

### 2.3. Satellite Data

2.3.1. Sentinel Data and Preprocessing

Because the in situ plot data were mainly collected in May 2017, we acquired one Sentinel-1 image (accessed on 20 May 2017) and two cloud-free Sentinel-2 patterns (accessed on 29 April 2017), the sensing periods for which were May and April 2017, respectively, from the European Space Agency Scientific Data Hub (ESA, https://scihub.copernicus.eu/ (accessed on 20 May 2017)). The atmospheric correction method was carried out using Sen2Cor v2.9 to transform the S2 TOA products into BOA products. For the SAR image, orbit calibration, radiometric calibration, Refined Lee (speckle) Filtering [43], terrain correction based on a 30 m resolution Shuttle Radar Topography Mission (SRTM) digital elevation model (DEM), and pixel value conversion to backscattering on the basis of Equation (1) were implemented:

$$\sigma^\circ(dB) = 10 \times lg\sigma^\circ \qquad (1)$$

where $\sigma^\circ$ is the normalized radar cross-section, and $\sigma^\circ$ dB is the converted value in decibels, as per convention. In addition, we acquired new S1 and S2 imagery by sustaining all other preprocessing in addition to the speckle filtering and the atmospheric correction to investigate the implications of speckle-filtered and unfiltered SAR products, S2 TOA images, and S2 BOA images. Additionally, all of the satellite images were based on map projection UTM Zone 50, WGS84, and the pixel values on the plots have been averaged and extracted using ArcGIS v10.7 (Table 2).

**Table 2.** Summary of Sentinel-2 and Sentinel-1 variables for forest AGB modeling.

| Sensor | Band/Index | Definition |
|---|---|---|
| Sentinel-2 MultiSpectral Instrument (MSI) | Band 2 | Blue, 490 nm, 10 m |
| | Band 3 | Green, 560 nm, 10 m |
| | Band 4 | Red, 665 nm, 10 m |
| | Band 5 | Red Edge, 705 nm, 20 m |
| | Band 6 | Red Edge, 749 nm, 20 m |
| | Band 7 | Red Edge, 783 nm, 20 m |
| | Band 8 | Near-infrared, 842 nm, 10 m |
| | Band 11 | Shortwave infrared 1, 1610 nm, 20 m |
| | Band 12 | Shortwave infrared 2, 2190 nm, 20 m |
| Sentinel-1 Synthetic Aperture Radar (SAR) | VV | Vertically transmitted and vertically received |
| | VH | Vertically transmitted and horizontally received |

### 2.3.2. Texture Measures

For all of the test plots, three first-order (occurrence) and three second-order (co-occurrence) S1 SAR dual-polarization textural metrics were adopted in our study. The three first-order textural measures included mean, data range, and variance, while the three second-order texture metrics were contrast, entropy, and correlation, which, respectively, belong to three main groups: the contrast group, the orderliness group, and the descriptive statistics group, according to [44]. Furthermore, all of the related textural features were computed from three sliding-window sizes: $5 \times 5$, $17 \times 17$, and $31 \times 31$. The textural measures were computed using ENVI (v5.3), using the formulas listed in Table 3.

**Table 3.** Textural features derived from Sentinel-1 dual polarization.

| Metric Type | Textural Metric | Formula |
| --- | --- | --- |
| First-order | Mean (ME) | $\frac{\sum_{k=1}^{N} x_k}{N}$ <br> where $x_k$ represents the gray-tone values of pixel k, and N represents the number of gray-tone values. |
| | Data range (RA) | $\max\{X\} - \min\{X\}$ <br> where X represents $x_1, x_2, \ldots, x_k$. |
| | Variance (VA) | $\frac{\sum_{k=1}^{N}(x_k - \bar{x})}{N-1}$ |
| Second-order | Contrast (CON) | $\sum_{n=0}^{N=1} n^2 \left\{ \sum_{i=1}^{N} \sum_{j=1}^{N} p(i,j) \right\}$ <br> where $n = \lvert i - j \rvert$. |
| | Entropy (ENT) | $-\sum_{i} \sum_{j} p(i,j) \log(p(i,j))$ |
| | Correlation (COR) | $\frac{\sum_i \sum_j (ij) p(i,j) - \mu_x \mu_y}{\sigma_x \sigma_y}$ <br> where $\mu_x$, $\mu_y$, $\sigma_x$, and $\sigma_y$ are the means and standard deviations of $p_x$ and $p_y$, where $p_x$ and $p_y$ are the marginal probabilities of x and y in the normalized GLCM. |

### 2.4. Forest AGB Prediction

First, we established univariate models through linear regression. In addition, every remotely sensed feature, which included S1 dual polarization, S2 spectral bands, and SAR-based texture measures, whether or not it was used in atmospheric correction and speckle filtering, was individually used to establish a univariate model for predicting the AGB of the oak, Chinese fir, Masson pine, and all of the plots, respectively.

Furthermore, based on different forms of preprocessing, feature sets using different combinations of S1 and S2 predictors were built using multiple linear regression (MLR). The multiple linear regression (MLR) model can be described with Equation (2):

$$y = a_0 + a_1 x_1 + \cdots + a_n x_n + \varepsilon \tag{2}$$

where $a_0$ is a constant; $x_1, \ldots, x_n$ are the explanatory variables; $a_1, \ldots, a_n$ represent the regression parameters related to the corresponding features; $y$ represents the value of the plot's forest AGB; n is the number of modeled variables; and $\varepsilon$ represents the error term. In addition, the relationships between the speckle-filtered and unfiltered TOA and BOA products were also detected.

Next, in order to compare the SAR-based textural measures with raw SAR dual polarization, we used feature importance in extreme gradient boosting (XGBoost), and selected the two most significant features among the SAR-based textures to model the AGB for the four kinds of plots. XGBoost is a highly efficient, end-to-end, and scalable boosting algorithm that employs a decision tree as its primary unit [45]. In comparison with the general gradient boost, it prevented over-fitting according to the regularized greedy forest, and it explored out-of-core computation and cache-aware learning, which have seldomly

been addressed [45]. On account of its outstanding sparse-data processing competence, which greatly escalates the algorithmic speed and reduces the computational memory required for large-scale data training, XGBoost has recently attracted a lot of attention [46]. In addition, it has been widely acclaimed as a means of modeling forest parameters [46,47]. In this study, n_estimators, max_depth, and the minimum sum of instance weight needed in a child (min_child_weight) were tuned when creating an XGBoost model for the forest AGB. The hyperparameters of XGBoost were tuned according to Grid CV in the scikit-learn package in Python v3.8, which is a grid search with MSE evaluation implemented through 10-fold cross-validation searching of the optimal groups of hyperparameters for the RF and XGBoost models. MLR models were also established to predict the AGB, to avoid algorithm dependencies. All of the models were built using Python v3.8 and Anaconda v3.

Eventually, 16 feature sets were established to investigate different combinations of S1 and S2 data on the basis of different preprocessing techniques. In the same measure, XGBoost and MLR were both used for modeling the forest AGB of four classes of plots. Figure 2 shows an overview of the steps taken for forest AGB prediction, while Table 4 lists the details of the 16 feature sets.

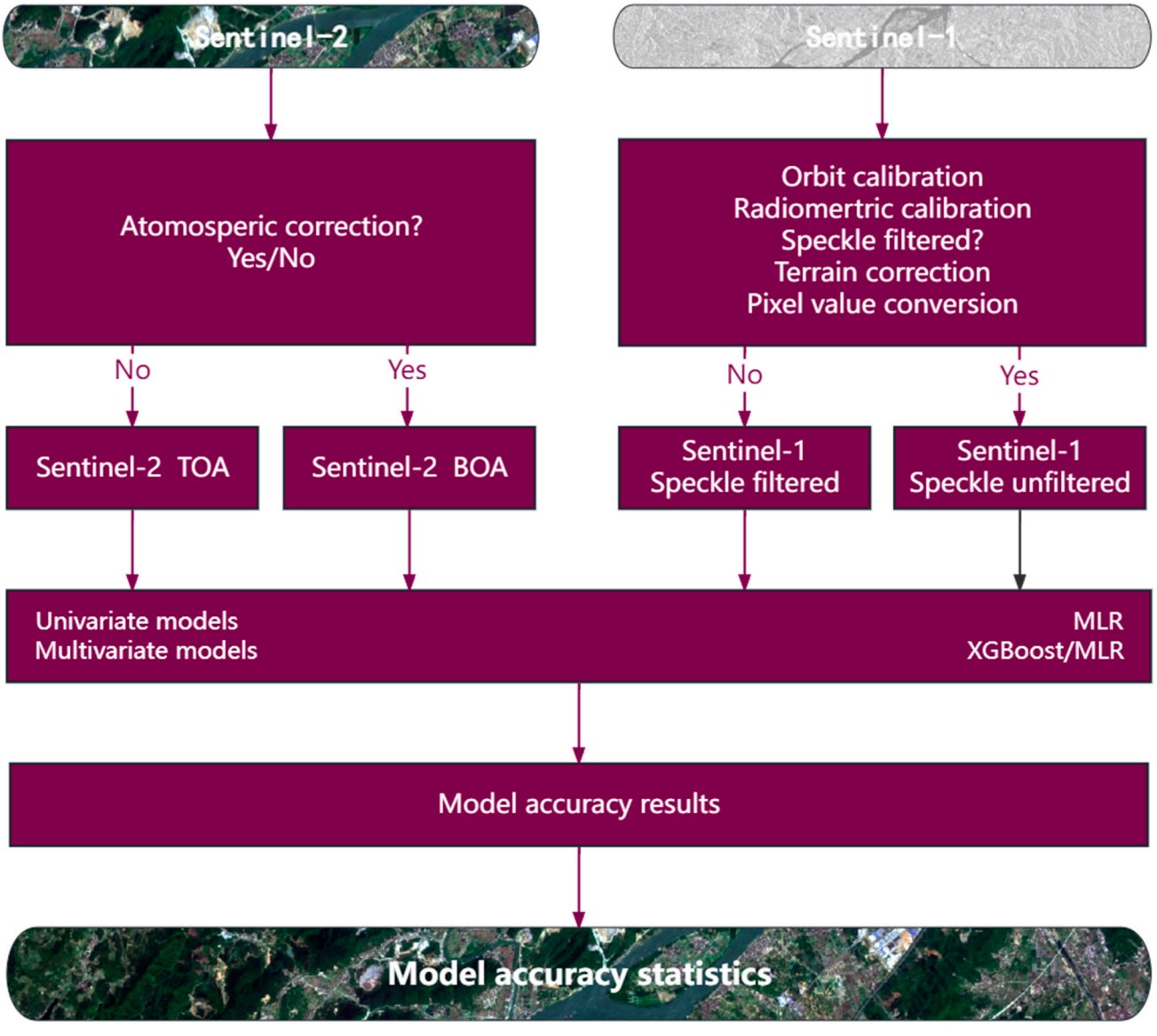

**Figure 2.** Overview of steps for forest AGB prediction.

Table 4. Feature sets established for modeling forest AGB.

| Feature Sets | Definition |
| --- | --- |
| A: $S2_{TOA}$ | S2 bands based on TOA product |
| B: $S2_{BOA}$ | S2 bands based on BOA product |
| C: $SAR/First/Second_{NOSPK}$ | Unfiltered SAR and all SAR-based textures |
| D: $SAR/First/Second_{SPK}$ | Speckle-filtered SAR and all SAR-based textures |
| E: $S2_{TOA} + SAR/First_{NOSPK}$ | Combines S2 TOA product, unfiltered SAR, and SAR-based first-order textures |
| F: $S2_{TOA} + SAR/First_{SPK}$ | Combines S2 TOA product, speckle-filtered SAR, and SAR-based first-order textures |
| G: $S2_{BOA} + SAR/First_{NOSPK}$ | Combines S2 BOA product, unfiltered SAR, and SAR-based first-order textures |
| H: $S2_{BOA} + SAR/First_{SPK}$ | Combines S2 BOA product, speckle-filtered SAR, and SAR-based first-order textures |
| I: $S2_{TOA} + SAR/Second_{NOSPK}$ | Combines S2 TOA product, unfiltered SAR, and SAR-based second-order textures |
| J: $S2_{TOA} + SAR/Second_{SPK}$ | Combines S2 TOA product, speckle-filtered SAR, and SAR-based second-order textures |
| K: $S2_{BOA} + SAR/Second_{NOSPK}$ | Combines S2 BOA product, unfiltered SAR, and SAR-based second-order textures |
| L: $S2_{BOA} + SAR/Second_{SPK}$ | Combines S2 BOA product, speckle-filtered SAR, and SAR-based second-order textures |
| M: $S2_{TOA} + SAR/First/Second_{NOSPK}$ | Combines S2 TOA product, unfiltered SAR, and all SAR-based textures |
| N: $S2_{TOA} + SAR/First/Second_{SPK}$ | Combines S2 TOA product, speckle-filtered SAR, and all SAR-based textures |
| O: $S2_{BOA} + SAR/First/Second_{NOSPK}$ | Combines S2 BOA product, unfiltered SAR, and all SAR-based textures |
| P: $S2_{BOA} + SAR/First/Second_{SPK}$ | Combines S2 BOA product, speckle-filtered SAR, and all SAR-based textures |

### 2.5. Statistical Methods for Predicting Forest AGB

In our work, the training samples, as well as the test samples, were randomly extracted, making use of the 10-fold cross-validation method for all experimental AGB models. The relationships and the accuracies between the predictors and the predicted forest AGB were assessed using the coefficient of association (R), based on Spearman's tests, the root-mean-square error (RMSE), and the relative RMSE (rRMSE), which were calculated according to Equations (3) and (4):

$$RMSE = \sqrt{\frac{1}{N}\sum_{i=1}^{N}(y_i - \hat{y}_i)^2} \tag{3}$$

$$rRMSE = \frac{RMSE}{\overline{y}} \times 100\% \tag{4}$$

where $y_i$ is the observed AGB value and $\overline{y}$ is the mean of observed AGB value; $\hat{y}_i$ represents the estimated AGB values and N is the number of test field plots.

### 3. Results

#### 3.1. Relationships between Different Preprocessing S1 Images and AGB

Figure 3 illustrates the relationships between all of the plots, the speckle-filtered images, and the unfiltered images from S1 SAR. All of the relationships between the two parameters were nearly zero, which shows a weak association between forest AGB and S1 dual polarization. Furthermore, the difference between the speckle-filtered and unfiltered images was also rather close to zero. Figure 4 portrays the distribution of the S1 VV polarization using different preprocessing methods. The distribution between the speckle-filtered and unfiltered S1 VV imagery was almost in superposition.

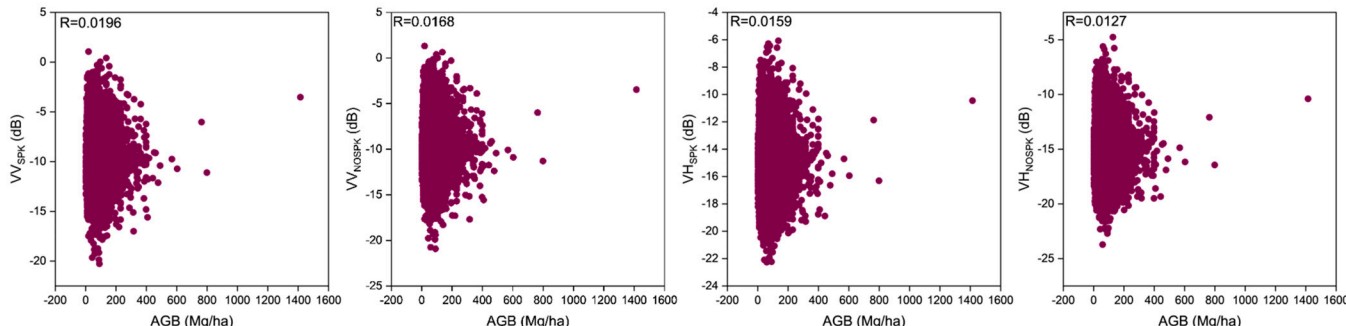

**Figure 3.** Scatter plots of forest AGB for all plots and for S1 dual polarization, with and without speckle-filtered imagery.

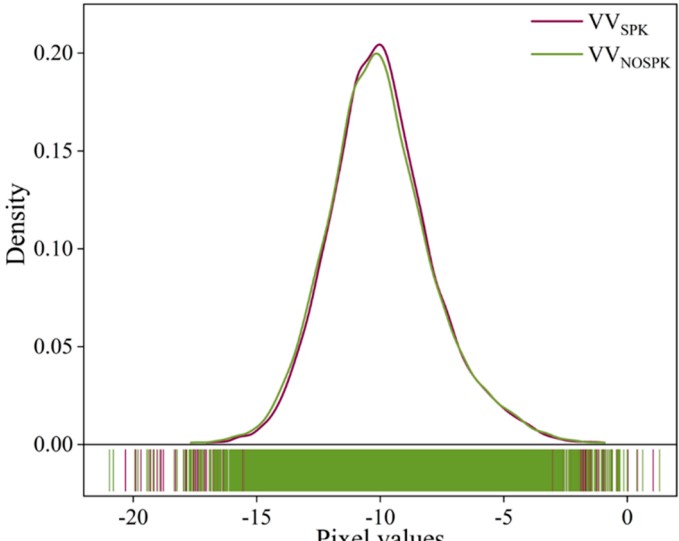

**Figure 4.** The distribution of SAR data based on VV polarization: speckle filtered vs. unfiltered.

### 3.2. S2 TOA and BOA Products for Modeling AGB

Figure 5 exhibits the performance of the S2 spectral bands derived from the studied imagery, with or without atmospheric correction. In the comparisons of the performances using the two kinds of S2 products, the accuracies we reached were pretty close. For oak, Chinese fir, Masson pine, and all of the plots, the highest accuracies in TOA were acquired, respectively, by B11 (rRMSE = 70.57%); B3 (rRMSE = 49.25%); B2 and B3 (rRMSE = 37.95%); and B7 and B8 (rRMSE = 66.09%), while the worst performances were demonstrated by B4 (rRMSE = 72.25%); B4 (rRMSE = 50.86%); B12 (rRMSE = 38.37%); and B4 (rRMSE = 69.04%), respectively. In contrast, the best-performing variables in the BOA products were, respectively, B11 (rRMSE = 70.55%); B3 (rRMSE = 49.26%); B2 and B3 (rRMSE = 38.06%); and B7 (rRMSE = 66.27%), while the worst were B4 (rRMSE = 72.31%); B4 (rRMSE = 50.93%); B12 (rRMSE = 38.37%); and B4 (rRMSE = 69.05%).

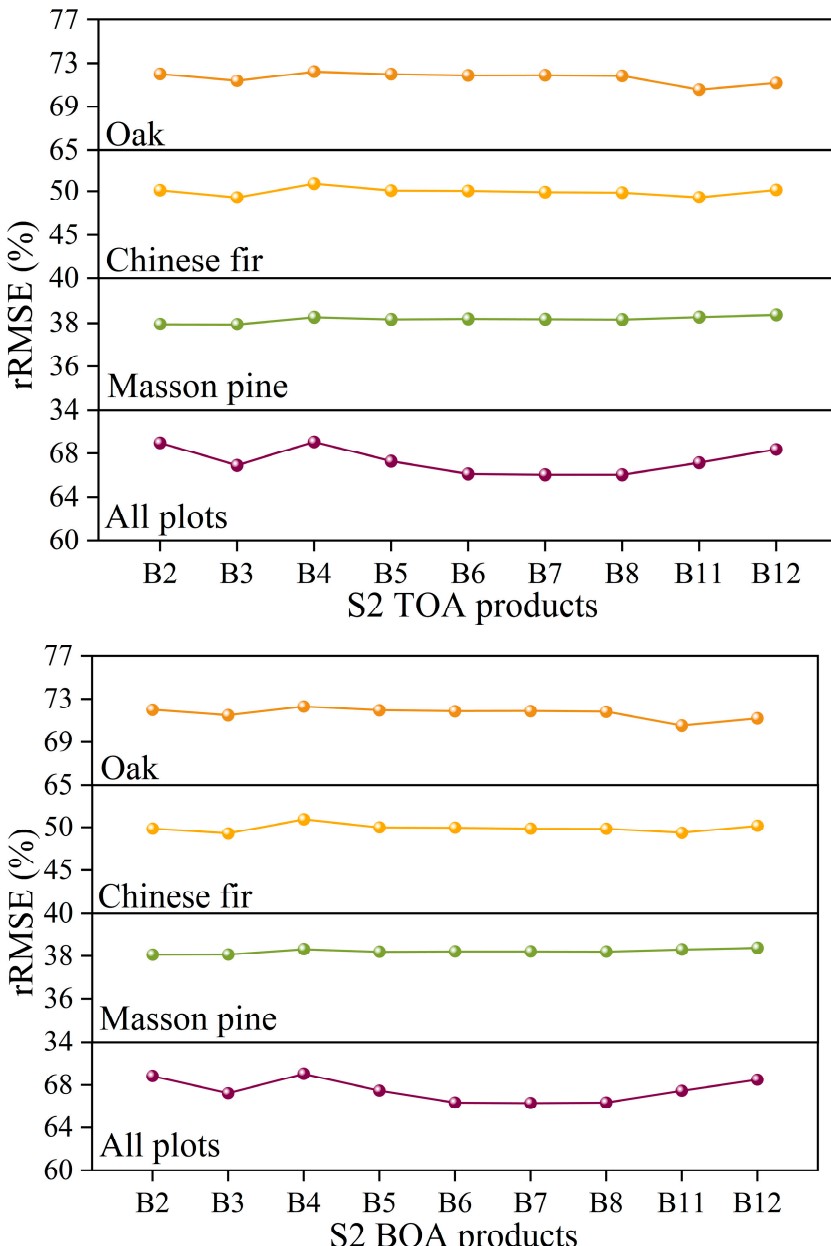

**Figure 5.** Performance of spectral bands based on S2 TOA and BOA products.

### 3.3. S1 SAR and Image Textures Using Different Preprocessing Techniques for Modeling AGB

#### 3.3.1. Univariate SAR Models

Table 5 delineates the predicted results using the polarized backscatter of S1, while Tables A1 and A2 show the performances of the first-order and second-order image textures for predicting AGB, respectively. Based on the results of the double polarizations, the performance was also very close for all kinds of plots. Tables A1 and A2 demonstrate that the best performance results between the first-order and second-order outcomes were adjacent. For all univariate predictors, the best predictors were $VV\_RA31_{NOSPK}$ (RMSE = 40.23 Mg/ha, rRMSE = 72.39%); $VV\_RA17_{NOSPK}$ and $VV\_ENT17_{NOSPK}$ (RMSE = 24.87 Mg/ha, rRMSE = 51.52%); $VH\_RA17$ (RMSE = 48.66 Mg/ha, rRMSE = 38.37%); and $VV\_CON31_{SPK}$ (RMSE = 52.42 Mg/ha, rRMSE = 69.03%) for oak, Chinese fir, Masson pine, and all plots, respectively, while the worst were, respectively, $VV\_ME31_{SPK}$ and $VV\_ME31_{NOSPK}$ (rRMSE = 72.76%); VV polarization backscatter based on speckle filtered, unfiltered, and all sliding-window sizes of $VV\_ME_{NOSPK}$ and $VV\_ME_{SPK}$

(rRMSE = 51.67%); speckle-filtered and unfiltered VV and VH polarization and numbers of SAR-based textures (rRMSE = 38.40%); and unfiltered VH and numerous SAR-based textures (rRMSE = 69.08%). The correlation coefficients (R) between the predictors, including all of the SAR-based data and the response AGB, were close to 0, indicating weak relationships. Despite the use or lack of speckle filtering, and despite the processing window size, neither the univariate models' performances between VV and VH polarization backscatter, nor the textural features derived from them, had any obvious difference. Actually, the best and worst performing predictors were also rather close. In addition, from Tables A1 and A2, it can be observed that the moving-window sizes did not exhibit regular performances. An interesting observation was that the use of textural measures was the best predictor for the four kinds of plots. In addition, the best- and worst-performing tree species were Masson pine and oak, respectively.

**Table 5.** Polarized backscatter of SAR for modeling forest AGB.

| Tree Species | SAR Bands | R | RMSE (Mg/ha) | rRMSE (%) |
|---|---|---|---|---|
| Oak | $VV_{NOSPK}$ | −0.038 ** | 40.42 | 72.74 |
| | $VH_{NOSPK}$ | 0.042 ** | 40.41 | 72.72 |
| | $VV_{SPK}$ | −0.043 ** | 40.42 | 72.75 |
| | $VH_{SPK}$ | 0.039 ** | 40.41 | 72.72 |
| Chinese fir | $VV_{NOSPK}$ | 0.014 | 24.94 | 51.67 |
| | $VH_{NOSPK}$ | 0.025 * | 24.93 | 51.65 |
| | $VV_{SPK}$ | 0.011 | 24.94 | 51.67 |
| | $VH_{SPK}$ | 0.025 * | 24.93 | 51.65 |
| Masson pine | $VV_{NOSPK}$ | −0.034 ** | 48.69 | 38.40 |
| | $VH_{NOSPK}$ | −0.036 ** | 48.69 | 38.40 |
| | $VV_{SPK}$ | −0.033 ** | 48.69 | 38.40 |
| | $VH_{SPK}$ | −0.034 ** | 48.69 | 38.40 |
| All plots | $VV_{NOSPK}$ | 0.010 | 52.45 | 69.07 |
| | $VH_{NOSPK}$ | 0.003 | 52.46 | 69.08 |
| | $VV_{SPK}$ | 0.014 | 52.45 | 69.07 |
| | $VH_{SPK}$ | 0.007 | 52.45 | 69.07 |

Note: R represents the relationship between the predictors and the AGB using Spearman's test. * Correlation is significant at the 0.05 level (2-tailed). ** Correlation is significant at the 0.01 level (2-tailed).

### 3.3.2. Multivariate SAR Models

Table 6 illustrates the predicted results of the SAR dual polarizations and their derived image textures. Apart from oak, similar to the performance carried out by single predictors, the outcomes predicted by image textures were slightly superior to those from polarized backscatter. Furthermore, for all kinds of tree species, the parametric models generally outperformed the nonparametric models. In comparisons of the univariate models, the models making use of double predictors (see Table 6) did not demonstrate better performance than the models with single predictors, and their performances were even worse than some of the univariate models.

**Table 6.** Selected variables based on SAR were established for modeling forest AGB.

| Tree Species | Selected Variables | MLR | | XGBoost | |
|---|---|---|---|---|---|
| | | RMSE (Mg/ha) | rRMSE (%) | RMSE (Mg/ha) | rRMSE (%) |
| Oak | $VV_{NOSPK}$, $VH_{NOSPK}$ | 40.40 | 72.71 | 42.07 | 75.71 |
| | $VH\_CON5_{NOSPK}$, $VH\_ME31_{NOSPK}$ | 40.43 | 72.75 | 40.72 | 73.28 |
| | $VV_{SPK}$, $VH_{SPK}$ | 40.40 | 72.70 | 42.10 | 75.84 |
| | $VH\_ME17_{SPK}$, $VH\_ME31_{SPK}$ | 40.42 | 72.74 | 41.28 | 74.33 |
| Chinese fir | $VV_{NOSPK}$, $VH_{NOSPK}$ | 24.93 | 51.65 | 24.98 | 51.75 |
| | $VH\_RA31_{NOSPK}$, $VH\_COR31_{NOSPK}$ | 24.91 | 51.60 | 24.87 | 51.53 |

**Table 6.** *Cont.*

| Tree Species | Selected Variables | MLR | | XGBoost | |
| --- | --- | --- | --- | --- | --- |
| | | RMSE (Mg/ha) | rRMSE (%) | RMSE (Mg/ha) | rRMSE (%) |
| Masson pine | $VV_{SPK}$, $VH_{SPK}$ | 24.93 | 51.64 | 25.07 | 51.93 |
| | $VH\_CON31_{SPK}$, $VV\_CON31_{SPK}$ | 24.92 | 51.63 | 24.81 | 51.40 |
| | $VV_{NOSPK}$, $VH_{NOSPK}$ | 48.70 | 38.40 | 48.83 | 38.51 |
| | $VH\_COR17_{NOSPK}$, $VH\_ME31_{NOSPK}$ | 48.66 | 38.37 | 48.83 | 38.51 |
| | $VV_{SPK}$, $VH_{SPK}$ | 48.70 | 38.40 | 48.91 | 38.57 |
| | $VH\_COR17_{SPK}$, $VH\_COR5_{SPK}$ | 48.66 | 38.38 | 48.85 | 38.53 |
| All plots | $VV_{NOSPK}$, $VH_{NOSPK}$ | 52.46 | 69.08 | 52.64 | 69.33 |
| | $VH\_ENT17_{NOSPK}$, $VH\_ME17_{NOSPK}$ | 52.40 | 69.01 | 52.54 | 69.18 |
| | $VV_{SPK}$, $VH_{SPK}$ | 52.45 | 69.07 | 52.54 | 69.19 |
| | $VH\_RA5_{SPK}$, $VV\_ME31_{SPK}$ | 52.41 | 69.02 | 52.60 | 69.27 |

### 3.4. Combinations of S2 and SAR Images for Modeling AGB

#### 3.4.1. Different Preprocessing of S2, S1, and Two Classes of Image Textures

Figure 6 illustrates the results of the integrations of the TOA and BOA products of S2, coupled with images from S1, with and without speckle filtering, to predict AGB. Obviously, the best-performing models were based on the spectral bands of S2, despite the use of MLR or XGBoost. The best relative RMS errors were 68.37% (B and XGBoost), 47.06% (B and XGBoost), 37.33% (A and MLR), and 64.99% (A and XGBoost) for oak, Chinese fir, Masson pine, and all plots, respectively. In contrast, the worst performance was generally generated by the VV and VH dual polarizations, and the predicted accuracies included 73.85% (L and XGBoost), 51.50% (C and MLR), 38.58% (D and XGBoost), and 69.24% (C and XGBoost). It was clear that, in the comparisons of the predictive models based on the two predictors, shown in Table 6, the performance demonstrated by the models based on nine predictors (groups C and D) was better.

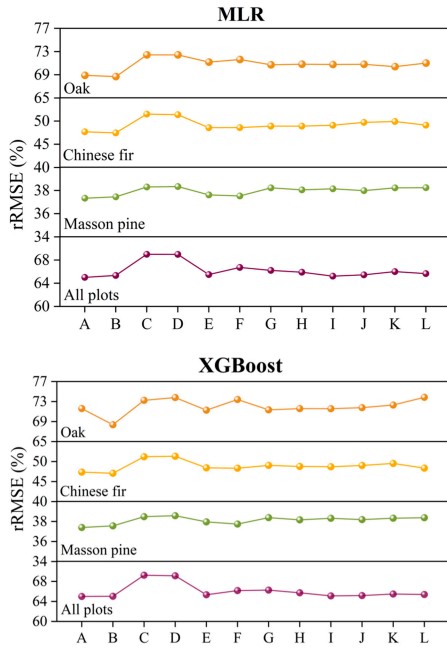

**Figure 6.** The predicted rRMSE (%) of the forest AGB for several kinds of plots using multivariate models implementing 12 groups of feature sets. The letters A–L represent the groups A–L as listed in Table 4.

Furthermore, for groups E–L, with the exception of oak, the top three best-performing groups for the four kinds of plots were, when using MLR, groups G, I, and K; E, F, and G; E, F, and K; and E, I, and J, respectively. Using XGBoost, they were E, G, and I; E, F, and L; E, F, and H; and E, I, and J. Although there was no conspicuously better performance between the first-order and second-order textural measures, most groups which achieved higher accuracies were based on the S2 TOA products. Additionally, the machine learning algorithm (XGBoost) did not show an evident enhancement when compared with MLR.

### 3.4.2. Different Preprocessing of S2 and S1

Figure 7 depicts the performances of groups M–P, which encompassed the synergies of utilizing S2 (TOA and BOA) products and S1 (speckle filtered and unfiltered) to predict the AGB. Figure 8 portrays scatterplots of the best-performing and worst-performing groups among the four kinds of plots, and the comparisons were implemented based on both statistical algorithms (XGBoost and MLR). For the four categories of plots, the best-performing groups included N (MLR, rRMSE = 70.89%); M (XGBoost, rRMSE = 48.39%); M (MLR, rRMSE = 38.09%); and M (XGBoost, rRMSE = 65.46%). In comparison, the lowest accuracies were achieved by groups M (XGBoost, rRMSE = 72.83%); N (MLR, rRMSE = 49.64%); P (XGBoost, rRMSE = 38.54%); and O (XGBoost, rRMSE = 66.30%), respectively. Among the four tree species, the worst-predicted AGB was that for oak, while the relative best was that for Masson pine.

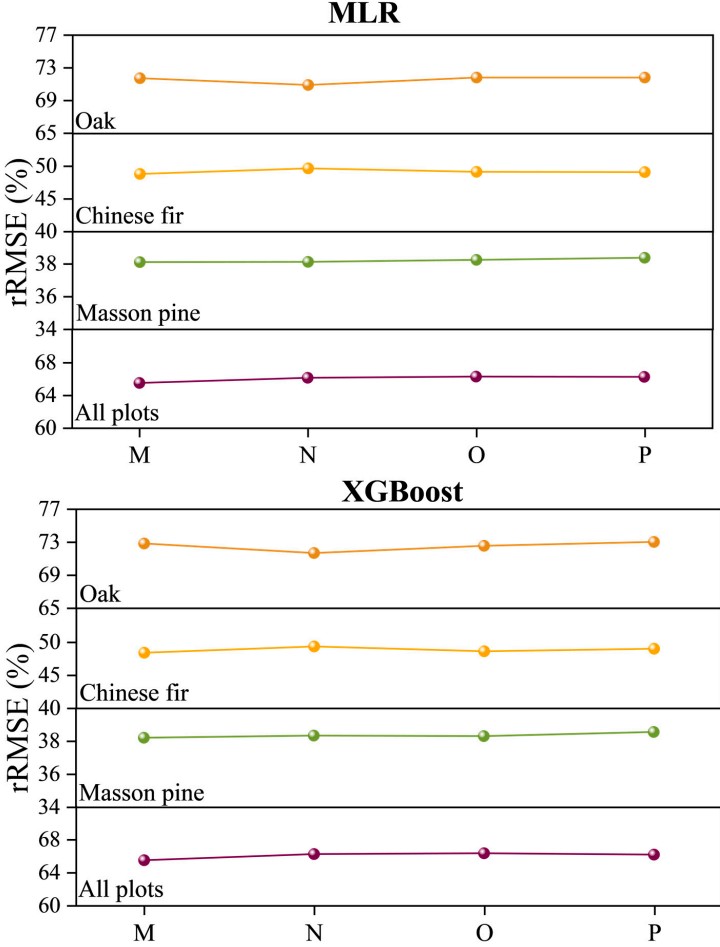

**Figure 7.** The predicted rRMSE (%) of the forest AGB for several kinds of plots using multivariate models implementing 12 groups of feature sets. The letters M–P represent the groups M–P as listed in Table 4.

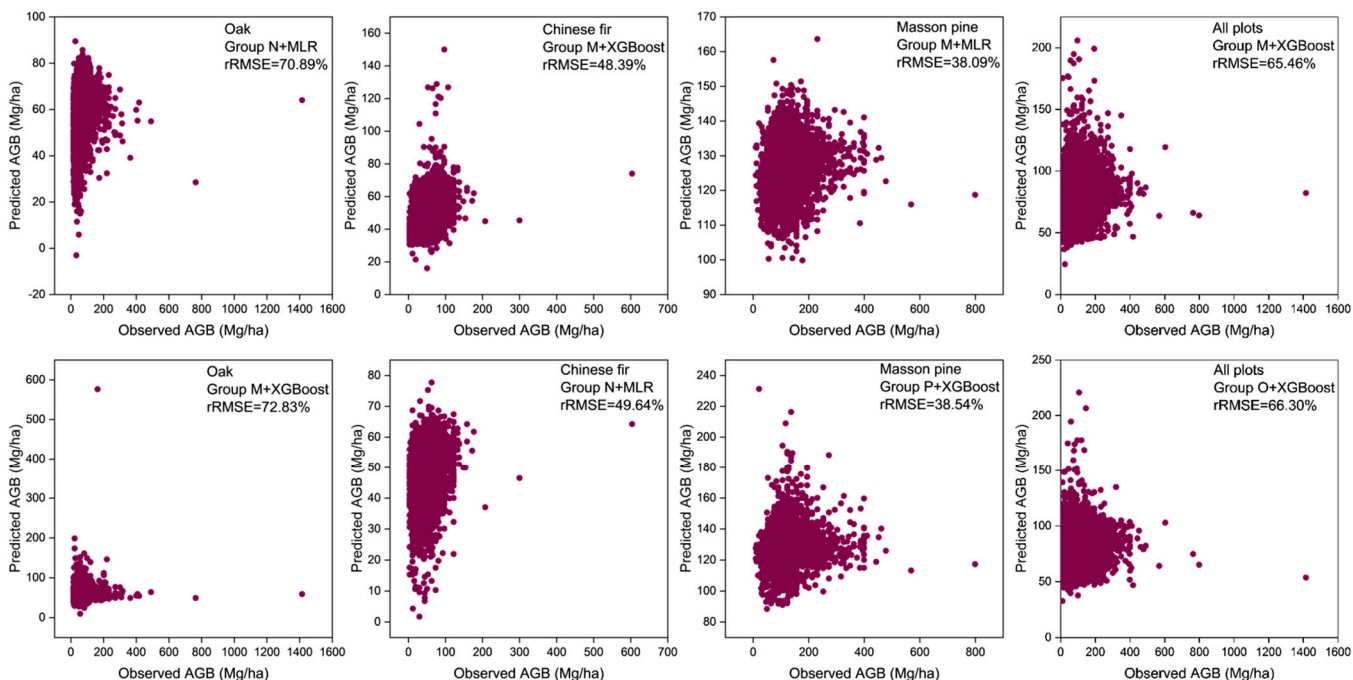

**Figure 8.** The scatterplots of the observed and the predicted AGB. The kinds of plots and the experimental groups are displayed in the upper right corner. Among groups M–P, the first row shows the worst-performing models, while the second row shows the best-performing models.

## 4. Discussion

Our goal in this research was to explore whether optical sensors, SAR, and their integration, utilizing disparate preprocessing methods, would make a great difference in predictions of forest AGB. Specifically, 16 groups of feature sets were established to model the AGB of four kinds of plots (oak, Masson pine, Chinese fir, and all plots) in Lin'an District, Zhejiang Province, China. In addition, among the feature sets, we also investigated the performances of combined first-order and second-order textural measures based on speckle-filtered and unfiltered S1 SAR images, and of TOA and BOA products based on S2 imagery.

### 4.1. S2 TOA and BOA Products

The generally similar results obtained via univariate spectral bands derived from TOA and BOA patterns demonstrated that this variation was responsible for a weak difference. Among the spectral bands, similar to the findings of previous research [48–50], the most crucial predictors included shortwave infrared 1 (SWIR 1/B11), Red Edge bands, and visible Green bands. For the SWIR bands, the SWIR spectrum is sensitive to canopy leaf water content in the forest stand structure [48]. Red Edge bands have been reported to have a high level of sensitivity in capturing information about the chlorophyll content of green plants [51]. We agree with the viewpoint that the Green band played a crucial role in the forest parameters found by [50]. We noted that the visible Blue band also reached the highest accuracies for Masson pine. Compared with the plots dominated by a single species, the traditional NIR band seemed to have more competence in summarizing heterogeneous areas (in all plots). On the other hand, the visible Red band (B4) demonstrated the worst performance in estimating the AGB for almost all kinds of plots (see Figure 5).

For multivariate S2 models, nine spectral bands were established for predicting forest AGB, and the S2 spectral bands showed the best performance among all of the feature sets. Consistent with univariate S2 spectral bands, the S2 TOA and BOA imagery did not demonstrate distinctly disparate levels of performance among any of the four kinds of forest AGB. The results show that the implications of atmospheric correction were minor for our survey plots, which is in line with what reference [52] revealed.

### 4.2. SAR and SAR-Based Texture Measures

In comparison with the unfiltered results, the results of the SAR images employing speckle filtering were almost identical in terms of data distribution. The results exhibited a weak difference made by speckle filtering and could explain the minor influence on the related yielded texture measures based on speckle-filtered and unfiltered S1 imagery. The scatter plots in Figure 3, which show the association between the forest AGB and speckle-filtered and unfiltered SAR imagery, also exhibit a weak function of speckle filtering. For first- and second-order texture analysis, no one processing window size outperformed other window sizes, and there was no conspicuous difference brought about by using speckle filtering. Obviously, the influence of speckle reduction was minor in SAR and SAR-based textural measures in the prediction of AGB for fine land cover, which is similar to what reference [52] found. Furthermore, the general, predicted accuracies of first- and second-order textural measures were nearly equivalent.

In our research, we found that a multitude of SAR-based textural measures were superior to raw S1 SAR dual polarization for predicting forest AGB. The performance modeled by the two most important SAR-based textural measures (see Table 6) outstripped dual-polarization VV and VH backscatter models, a finding which is in line with previous research findings [30,53]. In addition, the textural measures generated from VH polarization backscatter occupied more positions among the top two crucial textural measure models. This could be attributed to polarized VH having more pronounced surface variation conditions [54,55]. Compared with the model that had no representative textures of second-order textural analysis, the data range of the first-order texture measures (see Table A1) provided the most highly ranked results among the best-performing univariate SAR models, perhaps due to a greater level of sensitivity to field noise [56] or gray-level image transformation [57].

### 4.3. Integrations of S2 and S1 SAR for Modeling AGB

None of the integrations of S2, S1 SAR, or the SAR-based textural measures outstripped a single optical sensor, which is analogous to what reference [58] reported in their research about determining the age of acacia plantations in north–central Vietnam. They found that the synergistic employment of S1 C-band SAR and S2 did not provide any improvement over single optical S2 imagery. Additionally, the C-band SAR has also been considered to have low penetration to capture forest biomass features [30], while L-band or P-band SAR might be better for AGB estimations [59].

As for SAR-based, first- and second-order textural measures, we did not note a significant difference between them based on the predicted outcomes. The first-order and second-order textural measures were generated according to different calculation methods of SAR-based image intensity [35,60]. Compared with previous studies [39,60], the capacities of the two types of textural measures for enhancing single optical images were deemed as marginal by our research. In addition, for optical imagery, the best-performing outcomes were mostly based on S2 TOA products. Because some BOA products also showed the best accuracies (oak: group K and MLR; Chinese fir: group L and XGBoost), we could not fully explain why atmospheric correction lowered the S2 TOA imagery's prediction capacity, but we could demonstrate that the implications of the atmospheric correction of reduced aerosol or water vapor [19] were few. As a matter of fact, atmospheric correction has been explored for its accuracy in mapping land cover in Europe [52], and that research similarly found that atmospheric correction was marginal in its contribution to the classification of land-cover types.

For the combinations of all SAR-based textures and S2 spectral bands, the performance did not outperform those groups which were integrated with S2, and they only employed a kind of image texture feature. This was not surprising on account of the nearly equal predictive competence of the first- and second-order textures and the uncertainty of the SAR-based variables in manufacturing a better prediction by synergizing satellite optical imagery [58,60,61].

As for MLR and XGBoost, unlike previous studies which reported that machine learning methods obviously outstripped MLR [62,63], the difference in performance between the two statistical algorithms was small. By contrast, dozens of XGBoost models were even worse than MLR (e.g., Table 6 and Figure 7). In our comparisons of univariate SAR predictors, the models based on two predictors did not improve the performance, despite the statistical approach that we applied. Only when the models exploited the nine most important predictors could the predicted accuracies improve evidently (see Figures 6 and 7). Consequently, we suppose that this was caused by the limited influence of the statistical methods on the competence of the selected feature variables in predicting forest AGB.

*4.4. Assessment of Modeling Forest AGB*

From the predicted models, the most difficult plots to predict, among the integrations of S2 and S1 SAR, could be ranked as oak (rRMSE = 68.37%–73.85%) > All plots (rRMSE = 64.99%–69.24%) > Chinese fir (rRMSE = 47.06%–51.50%) > Masson pine (rRMSE = 37.33%–38.58%). It was obvious that the approaches could not be directly applied to practical applications, especially for oak, whose predictive performance for ABG was far worse than in previous research, which attained a 26% relative RMSE [64]. Even though the best-performing species, Masson pine, achieved 37.33% relative RMS errors, compared with the 19.69% RMSE acquired by [65] in Gaofeng Forest Farm in Nanning, Guangxi Zhuang Autonomous Region, the level of precision left room for improvement. The reason our forecast for forest AGB performed poorly could be linked to the low wavelength and penetration of C-band SAR [66]. Furthermore, the limited predictors in S1 SAR and S2 should be highlighted.

Open-access satellites have been extensively utilized for forest research, the cost of which could be immensely reduced by using light detection and ranging (LiDAR) or unmanned aerial vehicles (UAV). The preprocessing steps might help to address the noise data yielded by raw imagery [39]. However, for forest AGB related to fine land-cover parameters, we suggest that the emphasis should be put on predictors coming from satellite-based variables. Furthermore, in this research, we only conducted our investigation under the "best" conditions possible with respect to atmospheric pollution: using cloud-free images. Consequently, utilizing more areas of study and varying the time period are suggestions for further research.

## 5. Conclusions

In this work, we found that speckle filtering and atmospheric correction marginally influence the capacity of S2 spectral bands, SAR dual-polarization backscatter, and SAR-based textural measures to predict AGB. The associations between S1 SAR speckle-filtered and unfiltered imagery and S2 TOA and BOA products were strong. Additionally, textural measures, especially VH-based textures, outperformed the raw SAR dual-polarization models. This might be attributed to the fact that the backscatter textural measures contained the majority of vegetation diversity and density [67]. Ultimately, the groups that encompassed solely S2 imagery were the best-performing groups among all the feature sets, while the groups that included only S1-based data generally performed the worst. This was probably because of the data saturation of SAR [68,69]. In other words, we found that integrating S1 and S2 did not contribute to an improvement. XGBoost did not outstrip MLR for modeling AGB, in contrast with the findings of previous studies.

In a nutshell, in comparison with abundant variables related to remotely sensed data, the implications yielded by preprocessing in our study were too minor to make a great difference in predicting forest AGB. The statistical methods also showed little effect on making AGB predictions. We also suggest that the future studies about preprocessing could be focused on predictors derived from L-band or higher-wavelength band data, or satellite-based vegetation indices. Last but not least, more areas of study and variations in the time period are suggestions for subsequent satellite preprocessing research.

**Author Contributions:** Conceptualization, G.F.; Methodology, G.F.; Resources, L.F. and X.Z.; Data curation, G.F.; Writing—original draft, G.F.; Writing—review & editing, L.F. and X.Z.; Project administration, H.Y., L.F. and X.Z.; Funding acquisition, L.F. and X.Z. All authors have read and agreed to the published version of the manuscript.

**Funding:** This study was financially supported by the National Natural Science Foundation of China (Grant No. 42001354), the Natural Science Foundation of Zhejiang Province (Grant No. LQ19D010011), and the Zhejiang Provincial Key Science and Technology Project (Grant No. 2018C02013).

**Data Availability Statement:** The satellite data was downloaded from ESA (https://scihub.copernicus.eu/).

**Conflicts of Interest:** The authors declare no conflict of interest.

**Appendix A**

**Table A1.** Model results estimated by SAR-based, first-order texture features for the AGB of four kinds of plots.

| Plots | SAR Bands | Data Range | | | Mean | | | Variance | | | RMSE (Mg/ha) | R |
|---|---|---|---|---|---|---|---|---|---|---|---|---|
| | | 5*5 | 17*17 | 31*31 | 5*5 | 17*17 | 31*31 | 5*5 | 17*17 | 31*31 | | |
| Oak | VV$_{NOSPK}$ | 72.50 | 72.44 | **72.39** | 72.74 | 72.75 | 72.76 | 72.53 | 72.57 | 72.57 | 40.23 | 0.104 ** |
| | VH$_{NOSPK}$ | 72.60 | **72.54** | 72.55 | 72.72 | 72.72 | 72.72 | 72.61 | 72.61 | 72.62 | 40.31 | 0.080 ** |
| | VV$_{SPK}$ | 72.53 | 72.49 | **72.49** | 72.75 | 72.75 | 72.76 | 72.57 | 72.61 | 72.62 | 40.28 | 0.092 ** |
| | VH$_{SPK}$ | 72.59 | **72.55** | 72.58 | 72.72 | 72.72 | 72.73 | 72.62 | 72.63 | 72.71 | 40.31 | 0.081 ** |
| Chinese fir | VV$_{NOSPK}$ | 51.54 | **51.52** | 51.55 | 51.67 | 51.67 | 51.67 | 51.56 | 51.61 | 51.64 | 24.87 | 0.062 ** |
| | VH$_{NOSPK}$ | 51.59 | **51.58** | 51.63 | 51.65 | 51.65 | 51.64 | 51.60 | 51.62 | 51.65 | 24.90 | 0.045 ** |
| | VV$_{SPK}$ | **51.54** | 51.57 | 51.59 | 51.67 | 51.67 | 51.67 | 51.58 | 51.64 | 51.66 | 24.88 | 0.062 ** |
| | VH$_{SPK}$ | **51.58** | 51.59 | 51.64 | 51.65 | 51.65 | 51.64 | 51.61 | 51.64 | 51.65 | 24.90 | 0.045 ** |
| Masson pine | VV$_{NOSPK}$ | **38.39** | 38.39 | 38.40 | 38.40 | 38.40 | 38.40 | 38.39 | 38.39 | 38.40 | 48.68 | −0.011 |
| | VH$_{NOSPK}$ | 38.39 | **38.37** | 38.39 | 38.40 | 38.39 | 38.39 | 38.39 | 38.39 | 38.40 | 48.66 | 0.010 |
| | VV$_{SPK}$ | **38.39** | 38.39 | 38.40 | 38.40 | 38.40 | 38.40 | 38.39 | 38.39 | 38.40 | 48.68 | −0.007 |
| | VH$_{SPK}$ | 38.39 | **38.39** | 38.40 | 38.40 | 38.39 | 38.39 | 38.39 | 38.39 | 38.40 | 48.68 | −0.008 |
| All plots | VV$_{NOSPK}$ | 69.07 | 69.06 | 69.04 | 69.07 | 69.05 | **69.04** | 69.07 | 69.06 | 69.05 | 52.43 | 0.025 ** |
| | VH$_{NOSPK}$ | 69.08 | 69.06 | 69.05 | 69.07 | 69.05 | **69.05** | 69.07 | 69.07 | 69.06 | 52.43 | 0.021 ** |
| | VV$_{SPK}$ | 69.06 | 69.05 | **69.04** | 69.07 | 69.05 | 69.04 | 69.06 | 69.06 | 69.05 | 52.43 | 0.038 ** |
| | VH$_{SPK}$ | 69.07 | 69.06 | 69.05 | 69.07 | 69.05 | **69.04** | 69.07 | 69.06 | 69.06 | 52.43 | 0.023 ** |

The results before RMSE are the rRMSE predicted by first-order textures based on different SAR polarization backscatter and three different window sizes. The best-performing textural predictors of nine results in every line based on rRMSE are in bold. Based on the best-performing predictors, the RMSE, the correlation coefficients (R) between the predictors, and AGB using Spearman tests (** Correlation is significant at the 0.01 level (2-tailed).) are displayed.

**Table A2.** Model results estimated by SAR-based second-order texture features for the AGB of four kinds of plots.

| Plots | SAR Bands | Contrast | | | Entropy | | | Correlation | | | RMSE (Mg/ha) | R |
|---|---|---|---|---|---|---|---|---|---|---|---|---|
| | | 5*5 | 17*17 | 31*31 | 5*5 | 17*17 | 31*31 | 5*5 | 17*17 | 31*31 | | |
| Oak | VV$_{NOSPK}$ | 72.73 | 72.71 | 72.69 | 72.61 | 72.56 | 72.54 | 72.48 | **72.45** | 72.48 | 40.26 | 0.115 ** |
| | VH$_{NOSPK}$ | 72.74 | 72.72 | 72.71 | 72.65 | 72.60 | 72.60 | 72.58 | **72.47** | 72.52 | 40.27 | 0.100 ** |
| | VV$_{SPK}$ | 72.73 | 72.72 | 72.70 | 72.61 | 72.58 | 72.58 | **72.43** | 72.47 | 72.53 | 40.25 | 0.114 ** |
| | VH$_{SPK}$ | 72.74 | 72.72 | 72.72 | 72.66 | 72.62 | 72.64 | 72.53 | **72.48** | 72.54 | 40.28 | 0.106 ** |
| Chinese fir | VV$_{NOSPK}$ | 51.64 | 51.63 | 51.62 | 51.55 | **51.52** | 51.53 | 51.56 | 51.54 | 51.56 | 24.87 | 0.062 ** |
| | VH$_{NOSPK}$ | 51.65 | 51.65 | 51.64 | 51.59 | 51.56 | 51.59 | 51.62 | **51.55** | 51.59 | 24.88 | 0.053 ** |
| | VV$_{SPK}$ | 51.64 | 51.63 | 51.63 | 51.55 | **51.54** | 51.55 | 51.56 | 51.55 | 51.60 | 24.87 | 0.057 ** |
| | VH$_{SPK}$ | 51.65 | 51.65 | 51.64 | 51.60 | 51.58 | 51.61 | 51.61 | **51.56** | 51.61 | 24.89 | 0.050 ** |
| Masson pine | VV$_{NOSPK}$ | 38.39 | 38.39 | 38.40 | **38.38** | 38.38 | 38.39 | 38.39 | 38.39 | 38.40 | 48.67 | −0.004 |
| | VH$_{NOSPK}$ | 38.39 | 38.40 | 38.40 | 38.39 | 38.38 | 38.38 | 38.39 | **38.38** | 38.38 | 48.66 | 0.010 |
| | VV$_{SPK}$ | 38.39 | 38.40 | 38.40 | **38.38** | 38.38 | 38.40 | 38.39 | 38.38 | 38.40 | 48.67 | 0.001 |
| | VH$_{SPK}$ | 38.39 | 38.40 | 38.40 | 38.39 | 38.39 | 38.38 | 38.39 | **38.38** | 38.39 | 48.66 | 0.012 |
| All | VV$_{NOSPK}$ | 69.07 | 69.07 | 69.05 | 69.08 | 69.06 | **69.05** | 69.08 | 69.07 | 69.07 | 52.43 | 0.027 ** |

**Table A2.** *Cont.*

| Plots | SAR Bands | Contrast | | | Entropy | | | Correlation | | | RMSE (Mg/ha) | R |
|-------|-----------|----------|----------|----------|----------|----------|----------|----------|----------|----------|--------------|---|
| | | 5*5 | 17*17 | 31*31 | 5*5 | 17*17 | 31*31 | 5*5 | 17*17 | 31*31 | | |
| plots | VH$_{NOSPK}$ | 69.06 | 69.06 | **69.05** | 69.08 | 69.07 | 69.05 | 69.08 | 69.08 | 69.07 | 52.43 | 0.032 ** |
| | VV$_{SPK}$ | 69.07 | 69.06 | **69.03** | 69.07 | 69.06 | 69.06 | 69.08 | 69.07 | 69.07 | 52.42 | 0.037 ** |
| | VH$_{SPK}$ | 69.07 | 69.06 | **69.05** | 69.08 | 69.07 | 69.06 | 69.08 | 69.07 | 69.07 | 52.44 | 0.028 ** |

The results before RMSE are the rRMSE predicted by second-order textures based on different SAR polarization backscatter and three different window sizes. The best-performing textural predictors of nine results in every line based on rRMSE are in bold. Based on the best-performing predictors, the RMSE, the correlation coefficients (R) between the predictors, and AGB using Spearman tests (** Correlation is significant at the 0.01 level (2-tailed).) are displayed.

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
