# Peer review of "Synergistic Use of Sentinel-1 and Sentinel-2 Based on Different Preprocessing for Predicting Forest Aboveground Biomass"

_forests, doi:10.3390/f14081615_

Round 1

Reviewer 1 Report (New Reviewer)

Interesting manuscript, despite the fact that the preprocessing result did not make a great difference in predicting AGB, the authors did a lot of work, put forward and tested an interesting hypothesis.

 Line 40 - Hereinafter in the text, list the source numbers inside brackets [1,2,3]

Line 101 – aera - area

Figure 1 – rotate longitude signs on 90o to make images more compact, provide ticks or coordinate grid.

Field data 2.2

Apparently, the authors did not conduct field observations, but used the observations of 2017 from the Zhejiang Forestry Department. In this case, you should not single out a separate chapter Field data, but simply refer to the data source and indicate it in the bibliographic list. Or you need to name the chapter Field data source. It is advisable to indicate in what way, method, in which software the boundaries of research plots were determined. It is advisable to indicate the distribution map of research plots in the study area.

Line 122 – provide the source and parameters of RS data

 Table 1 - Here and below, format the tables in accordance with the requirements of the journal (at least bold column labels)

 Line 247 - Place table directly after mention in text

Line 284 – begin chapter with text

You no need to use Appendixes in manuscript, put Figures and Table in text if they are important for readers, and remove them, if not.

 References

Pay attention to the sources formatting according to journal demands.

Author Response

Reviewer 1:  

Note: The lines and pages were based on the Word revised version.

Interesting manuscript, despite the fact that the preprocessing result did not make a great difference in predicting AGB, the authors did a lot of work, put forward and tested an interesting hypothesis.

 Line 40 - Hereinafter in the text, list the source numbers inside brackets [1,2,3]

AR: We appreciate for your kind review and suggestions. The cited reference formats have been revised.

Line 101 – aera – area

AR: We are sorry for our carelessness. It has been revised.

Figure 1 – rotate longitude signs on 90o to make images more compact, provide ticks or coordinate grid.

AR: The map has been redrawn. See Figure 1.

Field data 2.2

Apparently, the authors did not conduct field observations, but used the observations of 2017 from the Zhejiang Forestry Department. In this case, you should not single out a separate chapter Field data, but simply refer to the data source and indicate it in the bibliographic list. Or you need to name the chapter Field data source. It is advisable to indicate in what way, method, in which software the boundaries of research plots were determined. It is advisable to indicate the distribution map of research plots in the study area.

AR: Thank you for suggestions. We have revised the chapter name. And the needed information has been added. See line 116-117, line 123-127.

With the help of the intersection of kilometer networks referring to the topographic maps with 1:10,000 map scale and National Forest Inventory standards, the research plots were divided using Arcgis, according to terrain boundaries…

Line 122 – provide the source and parameters of RS data

AR: The needed RS data has been added. See line 123-127.

The survey data acquired in the field were based on stratified random sampling with the help of remotely sensed data with a spatial resolution greater than 2.5 m (most were on the basis of high-resolution unmanned aerial vehicles (UAV)).

 Table 1 - Here and below, format the tables in accordance with the requirements of the journal (at least bold column labels)

AR: Thak you for your suggestions. We have made the column labels bold.

 Line 247 - Place table directly after mention in text

AR: It has been revised.

Line 284 – begin chapter with text

AR: It has been revised. See the cleaned version manuscript.

You no need to use Appendixes in manuscript, put Figures and Table in text if they are important for readers, and remove them, if not.

AR: Thank you for your kind considerations and suggestions. The figures have been reformatted.

 References

Pay attention to the sources formatting according to journal demands.

AR: Thank you for your suggestions. The references have been reformatted.

Reviewer 2 Report (New Reviewer)

The manuscript proposal uses an adequate methodology because the product used (input variables of the biomass model) is not.

The results found in the proposed manuscript (high rRMSE values, for example) prove that the wavelength (C band) is not suitable for, since it interacts almost exclusively with leaves and thin branches (van der Sanden, 1997 p. 62), unable to correctly model the biomass of forest species, since a large part of it is found in the trunks (more than 90%), an interaction that occurs only with the use of the L and P band (Woodhouse, 2006 p. 144 - https://doi.org/10.1201/9781315272573).

In addition, in-phase data has not been used, which could generate polarimetric decompositions (Cloude and Pottier Dual decomposition, in this case) since the scientific literature states that these decompositions produce results that are related to the predominant scattering mechanism in the scene, which usually improves for biomass prediction (https://doi.org/10.3390/f14050941).

Furthermore, when using optical data, it is well known that some vegetation indices are related to biomass which were not applied either (see: https://doi.org/10.1080/10106040608542399; doi:10.3233/978-1- 60750-494-8-201) that would probably help in a better prediction of the biomass in a more accurate way.

More details can be seen in the manuscript comments (attached).

References:

van der Sanden, J. J. (1997). Radar remote sensing to support tropical forest management. Wageningen University and Research.

Author Response

Reviewer 2:  

Note: The lines and pages were based on the Word revised version.

The manuscript proposal uses an adequate methodology because the product used (input variables of the biomass model) is not.

The results found in the proposed manuscript (high rRMSE values, for example) prove that the wavelength (C band) is not suitable for, since it interacts almost exclusively with leaves and thin branches (van der Sanden, 1997 p. 62), unable to correctly model the biomass of forest species, since a large part of it is found in the trunks (more than 90%), an interaction that occurs only with the use of the L and P band (Woodhouse, 2006 p. 144 - https://doi.org/10.1201/9781315272573).

In addition, in-phase data has not been used, which could generate polarimetric decompositions (Cloude and Pottier Dual decomposition, in this case) since the scientific literature states that these decompositions produce results that are related to the predominant scattering mechanism in the scene, which usually improves for biomass prediction (https://doi.org/10.3390/f14050941).

Furthermore, when using optical data, it is well known that some vegetation indices are related to biomass which were not applied either (see: https://doi.org/10.1080/10106040608542399; doi:10.3233/978-1- 60750-494-8-201) that would probably help in a better prediction of the biomass in a more accurate way.

More details can be seen in the manuscript comments (attached).

References:

van der Sanden, J. J. (1997). Radar remote sensing to support tropical forest management. Wageningen University and Research.

AR: Thank you for your kind considerations and suggestions. We also considered that L band and P band SAR might be better performed in AGB estimations. And the in-phase data and vegetation indices were also considered in previous research in AGB estimations. However, considering the lack of our data and the objective in our research was to identify whether the speckle filtered and atmospheric correction could make a difference in AGB estimation, we used C band SAR and considered several different dominant species in our research. As for L band or P band SAR, and vegetation indices, we promised we would make a deep thinking in our subsequent work in forest parameter estimations. And we have proposed your suggestion in section 5 Conclusion, see line 488-490.

Be careful the C band has a low vertical penetration capacity, mainly in forests, due to this wavelength.

In some cases, this wavelenght can suffer with Heavy Rainfall (Read: https://journals.ametsoc.org/view/journals/apme/39/6/1520-0450_2000_039_0840_qopiaf_2.0.co_2.xml). Complete/correct this sentence part.

AR: Thank you for your suggestions. We have revised our expressions. See line 48.

Figure A1???

AR: We are sorry that we did not express clearly that figure A1 exhibited the distribution of SAR VV. And we have revised the manuscript, see line 233-235.

I see no need to perform univariate models, as in the scientific literature, multivariate models for biomass prediction in SAR images present the best results.

AR: Thank you for your suggestions. The univariate models were used to compare raw dual-polarization with SAR-based texture measures, and to compare the difference of the SAR backscatters with or without speckle filtering.

In this case * and ** were represented as the same thing. Whats the difference between both?

AR: We are sorry for our carelessness. **Correlation is significant at the 0.01 level (2-tailed). See line 286.

This figure only makes sense if the two types of processing are compared for each species (i.e, two distribution curves (MLR and XGBoost for each species).

AR: The figures we used were to exhibit if different dominant species were influenced evidently under different preprocessing, and to make sense if the combinations based on different preprocessing were appropriate to different dominant species.

According to the scientific literature, the C band interacts mainly with leaves and branches in the tree canopy. How does citation [59] and [60] relate to the calculation of the biomass of the species in your research?

2 - The generation of target decomposition techniques (using phase) in SAR images shows promising results in establisment of biomass models  (For example: https://doi.org/10.3390/f14050941).

AR: Thank you for your advice and recommendation about target decomposition techniques (using phase) in SAR images. We appreciated that your supported reference allowed us to think in a new angle. And we would study it in detail in our subsequent research work. The inappropriate citation has been revised. See line 408-410.

Additionally, the C-band SAR has also been considered having low penetration to capture forest biomass features, while L-band or P-band SAR might be better for AGB estimations.

References:

Lu, Q. Chen, G. Wang, L. Liu, G. Li, and E. Moran, “A survey of remote sensing-based aboveground biomass estimation methods in forest ecosystems,” Int. J. Digit. Earth, vol. 9, no. 1, pp. 63–105, Jan. 2016, doi: 10.1080/17538947.2014.990526.

da S. Narvaes, J. R. dos Santos, P. da C. Bispo, P. M. de A. Graça, U. S. Guimarães, and F. F. Gama, “Estimating Forest Above-Ground Biomass in Central Amazonia Using Polarimetric Attributes of ALOS/PALSAR Images,” Forests, vol. 14, no. 5, p. 941, May 2023, doi: 10.3390/f14050941.

This AGB forecast is linked by the wavelenght used.

AR: Thank you for your advice. We have added explanations, see line 451-453.

The reason our forecast for forest AGB performed poorly could be linked to the low wavelength and penetration of C-band SAR.

References:

J. van der Sanden and D. H. Hoekman, “Potential of Airborne Radar To Support the Assessment of Land Cover in a Tropical Rain Forest Environment,” Remote Sens. Environ., vol. 68, no. 1, pp. 26–40, Apr. 1999, doi: 10.1016/S0034-4257(98)00099-6.

Whats the reason to?

AR: Thank you for your questions. We attributed it to the backscatter textural measures contained the majority of vegetation diversity and density. See line 478-479.

References:

Chen, Y. Wang, C. Ren, B. Zhang, and Z. Wang, “Optimal combination of predictors and algorithms for forest above-ground biomass mapping from Sentinel and SRTM data,” Remote Sens., vol. 11, no. 4, pp. 1–20, 2019, doi: 10.3390/rs11040414.

Same as the previous comment!

AR: We attributed it to data saturation of SAR. See line 481-482.

References:

Nuthammachot, A. Askar, D. Stratoulias, and P. Wicaksono, “Combined use of Sentinel-1 and Sentinel-2 data for improving above-ground biomass estimation,” Geocarto Int., vol. 37, no. 2, pp. 366–376, Jan. 2022, doi: 10.1080/10106049.2020.1726507.

Berninger, S. Lohberger, M. Stängel, and F. Siegert, “SAR-Based Estimation of Above-Ground Biomass and Its Changes in Tropical Forests of Kalimantan Using L- and C-Band,” Remote Sens., vol. 10, no. 6, p. 831, May 2018, doi: 10.3390/rs10060831.

What kind of variables already listed in the literature could make a difference, even in the C band, for forest biomass prediction?

AR: We are sorry that we just wanted to express the difference using different preprocessing of S2 and S1 were marginal, and the predicted variables (e.g., S1 vs. S2) should be more emphasized. And we have revised our expressions, see line 488-490.

We also suggest that the future studies about preprocessing could be put on predictors derived from L-band or higher-wavelength bands data, or satellite-based vegetation indices.

Round 2

Reviewer 2 Report (New Reviewer)

I am pleased to report that all recommendations and observations were carefully considered and integrated into the proposed manuscript. As a result, the final product meets the high standards required for publication in the journal.

This manuscript is a resubmission of an earlier submission. The following is a list of the peer review reports and author responses from that submission.

Round 1

Reviewer 1 Report

The manuscript describes a methodology to use the integration of satellite optical and SAR data for the estimation of forest biomass. Big efforts were spent to evaluate different pre-processing techniques of satellite data.

The manuscript is affected by some minor and majour issues.

MINOR ISSUES

1) The plants in the forest plots used as test dataset are rather young, this is noticeable from a the low AGB shown in table 1. This must be stated in the text and maybe in the title, since you can use the low penetration capability of C-band backscatter for forest estimation only in sparse or low biomass plots. They looks more like an arboretum

2) Figure 1a needs to be at the top of the page

3) LINE 114: what is the highr resolution (2.5m) satellite data?

4) LINES 132- 141: are very badly written, i.e. the statement «where ? sigma ° ?? is the normalized radar cross section and ? sigma° is the backscatter for a specific polarization» the correct statement is where ? sigma ° is the normalized radar cross section and ? sigma°db is just converted in decibel as convention». Furtherly. The statement « And the pixel values were based on mean values of S1 and S2 related images extracted from every plot» the correct statement (I presume) is «the value of each pixel on a plot has been averaged» or «each plot is represented by the average of all the pixels»

5) Table 2: VV backscatter vertical transmitted and vertical received

6) Figure 2: Very poorquality

MAJOUR ISSUES

1) The comparison between reflectance at top and bottom of atmosphere is pointless, since we are observing an ecosystem that is at bottom of atmosphere thus atmospheric correction is needed.

At the same time (I presume from the text) since you averaged all the backscatter on each plot the speckle filter is not important since the average is mantained after filtration (but not the variance).

2) Figure 4 and Figure 5 are incomprehensible and Figure 5 is poor quality. What does the captions mean («The performance implemented… « ????). What is the rMSE related to?

3) Why didn›t you add a sensitivity analysis of each satellite's features? I mean a scatterplot between backscatter and AGB in-situ measured to verify whether there is a consistent sensitivity (correlation) between the two parameters?

4) big efforts were spent in describing satellite data preprocessing that could be avoided and focusing on the description of models implementation and the obtaining of the results that are very poor. I.E. Figure 6 has been provided without a serious explanation on how those results were obtained.

In light of my comments, I advise against publication of the manuscript

The paper Synergistic Use of Sentinel-1 and Sentinel-2 Based on Different Preprocessing for Forest Aboveground Biomass Prediction is written with good English, nevertheless there are many repetitions in the text.